# The impact of roadway conditions towards accident severity on federal roads in Malaysia

**Mohad Fedder Musa**[1⊚], **Sitti Asmah Hassan**[2⊚]*, **Nordiana Mashros**[2⊚]

**1** Public Work Department, JKR, Kuala Lumpur, Malaysia, **2** School of Civil Engineering, Faculty of Engineering, Universiti Teknologi Malaysia, Skudai, Johor, Malaysia

⊚ These authors contributed equally to this work.
* sasmah@utm.my

**Data Availability Statement:** All relevant data are within the manuscript and its Supporting Information files.

**Funding:** The authors are grateful to Universiti Teknologi Malaysia and Ministry of Education for

## Abstract

The fatal accidents on the roads remain a global concern. Daily, approximately 18 traffic accidents occur in the Peninsular Malaysia that cause on an average one death in every hour, a situation that needs preventive measures. The development of the effective strategies to reduce such fatal accidents requires the identification of various risk factors including the road condition. We identified such accident severity issues using the public work and police department databases that consisted of 1067 cases of various severity levels occurred on the Malaysian federal roads during 2008 to 2015. These records were used to develop ordered logistic regression model for the accident severity and nine variables were analyzed. The results revealed that the presence of poor horizontal alignment affected the model outcomes. The likelihood of the more serious accident severity due to the poor horizontal alignment was correspondingly about 0.4 times less compared to the absence of such factors. It is established that the present findings may assist the local authorities to take proactive actions to prevent serious road accidents on the road segments possessing the standard horizontal alignment.

## 1. Introduction

With the growing populations and vehicles on the roads, the traffic accidents and related deaths have been ever-increasing worldwide [1–3]. This alarming situation needs to be inhibited to circumvent the immense economic loss and save lives. According to the reports of the World Health Organization [4] and World Bank [5], the nations with low and middle earnings are severely affected by the fatalities of the road traffics, contributing approximately 90% of the road accidents related deaths globally. In fact, the population with working age ranged from 15 and 64 years carry the lion's share of the road accident fatalities and long-term disabilities. An estimate since 2000 revealed that, the road accidents in Malaysia alone is responsible for the increase of the death rate by approximately 14% [6]. According to the Malaysian Ministry of Transport [7] report, the total number of road accidents over the past decade have increased by approximately 41%, amounted to 369,319 and 521,466 mishaps in the year of 2007 and 2016, respectively. Consequently, the total number deaths were augmented from 6,282 to

the financial (Q.J130000.2522.17H71 and R.
J130000.7851.5F024) and technical assistance.

**Competing interests:** The authors have declared
that no competing interests exist.

7,152. In the same period, the total number of vehicles involved in the road accidents were increased from 666,027 to 960,569 (approximately 44% rise). In Malaysia, the private vehicles being the primary mode of transportation [8,9]. Its number (officially registered) has been steadily increased from 97,262 (in the year of 1980) to 284,461 (in the year of 2016) [10]. Fig 1 shows the number of registered vehicles in Malaysia according to their types.

Despite the enormous investment by the Malaysian Government to develop the road infrastructures the traffic management and road safety monitoring remains deficient. Therefore, the increased mobility of the current society impacted negatively on the road safety aspects. To reduce the rate of traffic accidents, several road safety programmes has been launched in the past. These include the accident prevention (proactive measures such as the road audits), accident reduction (reactive measures such as the improvements at the hazardous locations and infrastructure development targeted at the vulnerable road users for example the motorists), road maintenance (for example the grass cutting and potholes paving) and constructions of the new roads. In spite of all these proactive and reactive measures, both traffic congestion and number of fatal accidents in Malaysia are still reported to be higher compared to other nations [11–14], indicating the need of more proactive measures to lessen the numbers of road accidents. These road accidents negatively affect the national economy where the government need to allocate an extra budget to repair the broken infrastructures caused by the accidents. Such accidents also involve the administrative cost, medical compensation, loss of manpower and productivity [15–17].

Many factors are responsible for the road accidents and they are mostly related to the users (human error), environmental and vehicles [3,12,16,18–23]. The gender, age, experience, physical and behavioural characteristics of the drivers (for example the reaction time, risk taking, decision making and seat belt usage) can contribute significantly to the chance and severity of the accidents [24–30]. In addition, various environmental factors such as the weather and lighting conditions in a particular time of the day were shown to affect both likelihood and severity of an accident [3,19,25,27–29,31–32]. On top, various attributes of the vehicle such as its type, age, quality of the tyres, braking system, steering and safety features were proven to contribute to the risks of injury severity in an accident [19,27–28]. In a case study,

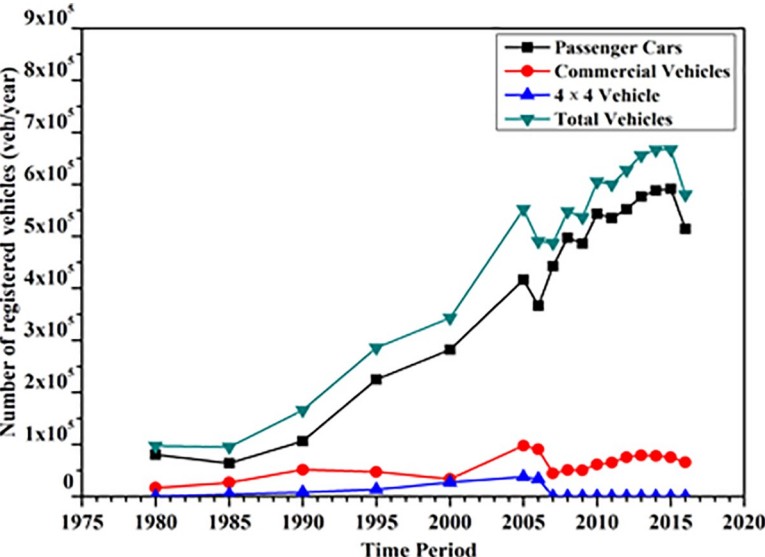

**Fig 1. The number of vehicles registered from 1980 to 2016 [10].**

Wedagama [33] has demonstrated that the collisions with the pedestrians and other vehicles could contribute to the fatal accidents of the motorcycle. The traffic and trip characteristics such as the traffic volume and composition together with the speed measurements can be considered as the common variables in many road accident models [26,34]. In the Northern province of Ghane, the overloading and obstruction related factors were discerned to contribute to the accident severity [35]. Yet again, the road characteristics including the median barriers, shoulders and lane widths, flat and perpendicular bends, number of access points and pavement conditions were shown to play different role on the probability and severity of the road accidents [3,29–30,34,36–39].

Over the years, various statistical approaches have been used for the analyses of road accidents [40–42]. To evaluate the impact of various factors on the accident severity, the celebrated logistic regression (LR) strategy was widely utilized [3,19–20,24,27,30,33,35,37,39,43–45]. In addition, the ordered logistic was also used in the diverse accident severity prediction models [46–47]. Generalized ordered logistic regression [48–50], latent segmentation based generalized ordered logistic regression [51], a hierarchical ordered logit model [52], random parameters logit model [53], random parameters multinomial logit model [54], mixed logit model approach [55–60], a latent class mixed logit model [61], standard OLS, the quantile regression and the extreme value theory [62], Markov-switching approach [63], ordered probit model [64] and random parameters bivariate ordered probit model [65] were also utilized in the accident prediction models. Several studies proposed Bayesian spatial generalized ordered logit model approach as a recent development in predicting accident severity [66,67]. Many attempts have been made to estimate the effectivessness of the general discrete-outcome approach in the accident severity models.

The general question related to the impact of various contributing factors to the accident severity needs further clarification. To provide some basic insight, present work evaluated the influence of the geometrical road factors on the accident severity chances in Malaysia. The effects of many other factors were neglected because of considerable limitations associated to the data obtained from the accident reports. The outputs of the developed ordered logistic regression model were analysed to signify the influence of the inconsistent road factors on the accident severity. The role that good traffic engineering and road design dealing with the ongoing accident problems were emphasized.

## 2. Methodology

In recent times, the accident mitigation strategies on the federal roads of Malaysia due to the presence of complex geometrical design standards and soaring traffic volumes became a priority. In this perception, we developed an accident severity ordered logistic regression model (OLRM) to evaluate the factors responsible for the accident severity on the Malaysian federal roads. A complete set of data concerning the roadway conditions were obtained from the databases of the Public Work Department (PWD) and police reports and utilized in the model for the prediction.

### 2.1 Data collection

The study was conducted at federal roads in Peninsular Malaysia. All such highways are declared under the Federal Roads Ordinance (1959). These are the main interurban roads connected to the state capitals, leading to the entry and exit points in the country.

The cited federal roads accidents' data that occurred during 2008 to 2015 were obtained from the PWD Forensic Accident Reports and Royal Malaysia Police (RMP) POL27 accident records database, which were integrated in the model for further analysis. Total 166 accident

**Table 1. The number of accidents and its percentages as a function of 9 variables (total number of accident reports, n = 166).**

| No. | Variables | Number (percentage) | |
|-----|-----------|------------------|------------------|
| | | **Good Conditions** | **Bad Conditions** |
| 1. | Signing and Marking (SM) | 33 (19.9%) | 133 (80.1%) |
| 2. | Road Design Consistency (RDC) | 50 (30.1%) | 116 (69.9%) |
| 3. | Traffic Barrier (TB) | 99 (59.6%) | 67 (40.4%) |
| 4. | Pavements (P) | 102 (61%) | 64 (39%) |
| 5. | Shoulders (S) | 99 (60%) | 67 (40%) |
| 6. | Lighting (L) | 138 (83%) | 28 (17%) |
| 7. | Sight Distance (SD) | 105 (63%) | 61 (37%) |
| 8. | Horizontal Alignment (HA) | 142 (85.5%) | 24 (14.5%) |
| 9. | Environmental Impact (EI) | 135 (81%) | 31 (19%) |

reports with 1067 cases (fatals, serious injuries and slight injuries) were analysed. Table 1 depicts the road conditions (also called variables with two attributes good and poor) related potential risk factors (number of accidents and percentages) that were identified. The good and poor characteristics of these variables were classified when they conformed to the PWD specification and not in conformity, respectively [68].

## 2.2 Development of ordered logistic regression model (OLRM)

In the proposed OLRM, the accident severities were used as the dependent variables. Conversely, the potential risk factors of the accident severities were used as independent variables. A total of nine (9) independent variables were modeled with accident severity as the dependent variable. Table 2 shows the description code of each variable.

To test the assumption of multicollinearity, bivariate correlation test was conducted on the independent variables. As a final test, the variance inflation factor (VIF) was used to evaluate correlation for independent variables. A VIF less than 10 is acceptable for most published literature [69–70]. Using the proportional odds test (Brant test), the assumption of parallel regression was tested and it was found that the assumption was not violated. Therefore,the ordered logistic regression was found to be appropriate for the study.

Tables 3 and 4 presents the ordered logit models for independent variables and intercepts. The variables kept in the model are those whose parameters are statistically significant at the 90% level.

**Table 2. The description of variables used in the model.**

| Variable | Code | Description of code |
|----------|------|---------------------|
| Dependent | 0 | No slight injury, serious injury or fatal |
| | 1 | Slight injury only |
| | 2 | Serious injury only |
| | 3 | Slight and serious injuries |
| | 4 | Fatal only |
| | 5 | Slight injury and fatal |
| | 6 | Serious injury and fatal |
| | 7 | Slight injury, serious injury and fatal |
| Independent | 0 | Good condition and conform to the PWD specification |
| | 1 | Poor condition and do not follow PWD specification |

**Table 3. Ordered logistic regression modelling results for independent variables.**

| Variable | Estimate | Std. error | Wald Chi Square | p-value | Odd ratio (OR) |
|---|---|---|---|---|---|
| RDC | 0.88134449 | 0.7920365 | 1.2452 | 2.66E-01 | 2.4140 |
| SD | 0.36610062 | 0.3370551 | 1.1893 | 2.77E-01 | 1.4421 |
| HA | -0.87500450 | 0.4633180 | 3.5791 | 5.90E-02 | 0.4169 |
| P | 0.27691086 | 0.3329731 | 0.6951 | 4.06E-01 | 1.3190 |
| S | -0.29268489 | 0.3225310 | 0.8237 | 3.64E-01 | 0.7462 |
| TB | -0.40864583 | 0.3380687 | 1.4631 | 2.27E-01 | 0.6646 |
| L | 0.09241950 | 0.4730424 | 0.0382 | 8.45E-01 | 1.0968 |
| SM | 0.42374739 | 0.4140275 | 1.0427 | 3.06E-01 | 1.5276 |
| EI | 0.02463085 | 0.3956738 | 0.0039 | 9.50E-01 | 1.0249 |

RDC: Road design consistency; SD: Sight distance; HA: Horizontal alignment; P: Pavements; S: Shoulders; TB: Traffic barrier; L: Lighting; SM: Signing and marking; EI: Environmental impact.

Based on both Tables 3 and 4, with event $i$ as the default, the probability that event $j$ could happen is:

$$P(i|j) = \frac{e^{\left[\begin{array}{c}0.88134449 * \text{RDC} + 0.36610062 * \text{SD} - 0.87500450 * \text{HA}+ \\ 0.27691086 * \text{P} - 0.29268489 * \text{S} - 0.40864583 * \text{TB}+ \\ 0.09241950\text{L} + 0.42374739 * \text{SM} + 0.02463085 * \text{EI} + \alpha_{ij}\end{array}\right]}}{1 + e^{\left[\begin{array}{c}0.88134449 * \text{RDC} + 0.36610062 * \text{SD} - 0.87500450 * \text{HA}+ \\ 0.27691086 * \text{P} - 0.29268489 * \text{S} - 0.40864583 * \text{TB}+ \\ 0.09241950\text{L} + 0.42374739 * \text{SM} + 0.02463085 * \text{EI} + \alpha_{ij}\end{array}\right]}}, \quad (1)$$

where $i = 0,\ldots,6$ and $j = 1,\ldots,7$.

## 3. Results and analysis

### 3.1 Accident trends

Fig 2 displays the accident trends on the Malaysian federal roads which consists of 1067 accident cases. The most reported accident type on the federal roads was identified as "out of control". Interestingly, most of the accidents occurred during good weather condition regardless of the day time. This situation occurred due to drivers' behaviour such as carelessness, reckless drivers, overspeeding.

**Table 4. Ordered logistic regression modelling results for intercepts.**

| i | j | Intercept estimate ($\alpha_{ij}$) | Std. error | t-value | p-value |
|---|---|---|---|---|---|
| 0 | 1 | -3.14008997 | 0.52872510 | -5.93898368 | 2.87E-09 |
| 1 | 2 | -2.97478823 | 0.50538170 | -5.88622032 | 3.95E-09 |
| 2 | 3 | -2.83063870 | 0.48701140 | -5.81226399 | 6.16E-09 |
| 3 | 4 | -2.83063843 | 0.48701140 | -5.81226343 | 6.16E-09 |
| 4 | 5 | -0.66563865 | 0.36335440 | -1.83192662 | 6.70E-02 |
| 5 | 6 | -0.29553911 | 0.36164440 | -0.81720925 | 4.14E-01 |
| 6 | 7 | 2.58596727 | 0.43949820 | 5.88390806 | 4.01E-09 |

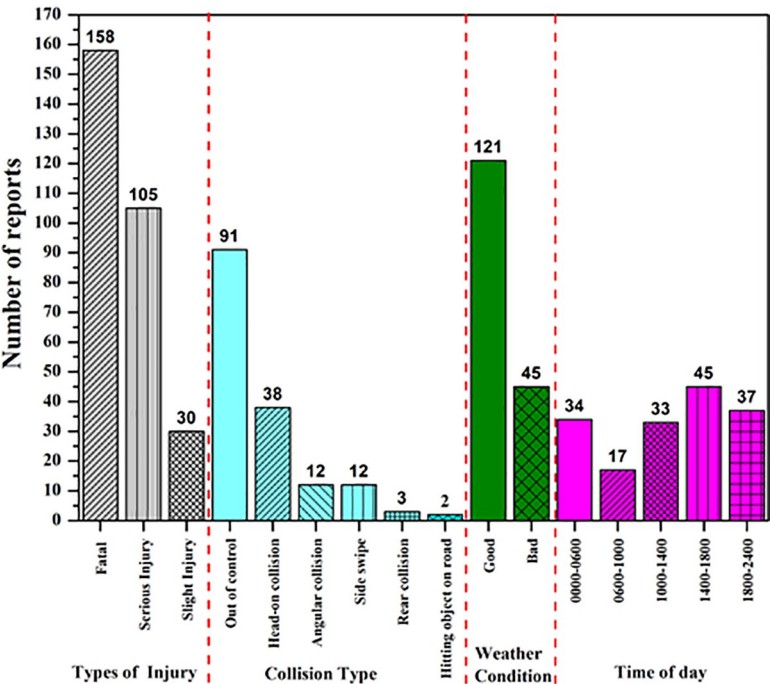

**Fig 2. Accident trends on the federal roads of Malaysia revealing the correlation among different variables.**

As shown in Fig 3, approximately 55% of the 1067 accident cases was found to involve in the fatalities. It was also observed that the cars were the major vehicles involved in the accidents (approximately 59%).

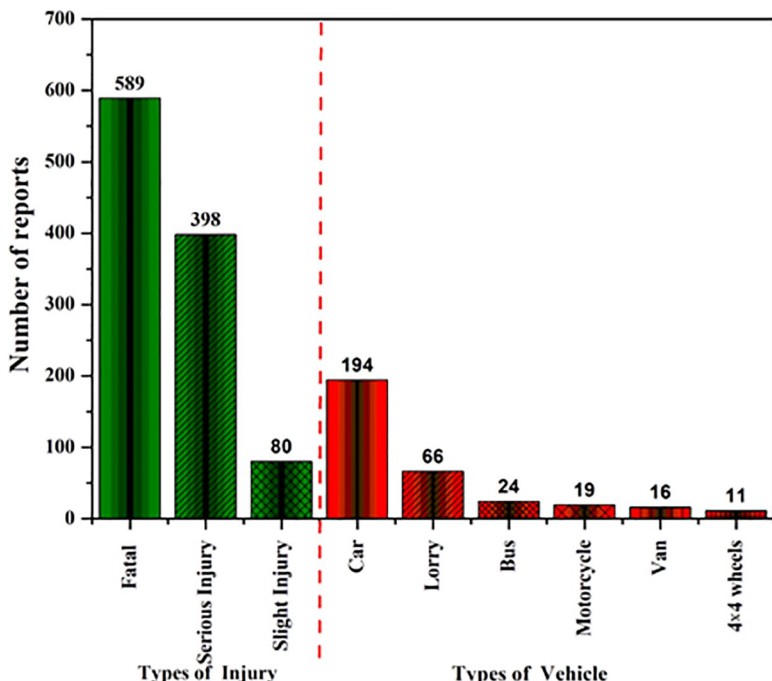

**Fig 3. Number of victims on accident cases involving people and vehicles on the federal roads of Malaysia.**

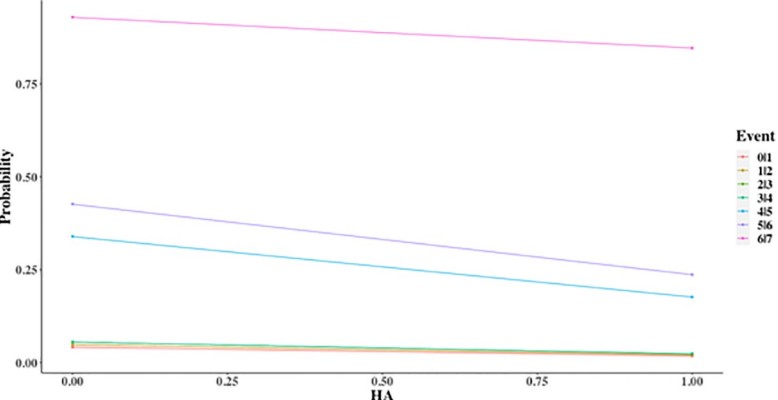

**Fig 4. Injury severity probability for horizontal alignment (HA).**

### 3.2 Potential risk factors

Table 3 summarizes the values of various coefficients obtained from the model. The potential risk factors for the fatal accidents consisting of nine variables were considered in the selection processes.

The causal factors that contributed significantly to the accident severities have also been identified. The achieved values of the odds ratio of 0.4169 for the horizontal alignment was less than 1, indicating that for every additional increment in the poor horizontal alignment the odds of this horizontal alignment contributed to more severe accident decreased by a factor of 0.4169. This in turn suggested that the poor horizontal alignment in federal roads was less likely to be responsible for the more serious accident severity. Table 3 shows that only horizontal alignment (HA) is significant towards the severity level of road accidents at 90% of confidence level (p-value = 0.0590 < 0.1). The relation of the significant variable in the ordered logistic regression model is followed:

$$P(i|j) = \frac{e^{-0.87500450*HA+\alpha_{ij}}}{1+e^{e^{-0.87500450*HA+\alpha_{ij}}}}, \tag{2}$$

where $i = 0,\ldots,6$ and $j = 1,\ldots,7$.

The probability of the accident severity level was decreased with the increase of horizontal alignment inconsistency (predicted as negative sign). The probability plot of injury severity based on horizontal alignment is as in Fig 4.

In general, Fig 4 shows the same probability trend across all events. The downward trend implies that the probability of a more serious event to happen decreases when the horizontal alignment is in a poor condition and does not follow PWD specification.

This communication depicts the development of accident severity model by considering the most significant geometrical road factors attributes to accident severity on federal roads in Malaysia with which is horizontal alignment. The modelling outcome suggested that such factors are greatly served the respective stakeholders in managing the road facilities.

## 4. Discussions and conclusions

The proposed model was simulated to predict the likelihood of the accident severity caused by diverse road factors so that some effective preventive measures can be taken to reduce such accidents. The projected model may pave the way to the relevant authorities to initiate preventive measures which is inexpensive rather than undertake costly corrective measures in long

run depending on the unfavourable road conditions that cause such serious accident severity. For instance, audible edge line treatments can be introduced in order to help road users become more alert, resulting in positive outcome in accident reduction as reported by [71].

The model results indicated that the presence of poorer condition in the horizontal alignment led to more lesser probability of severe injury during the accidents on the Malaysian federal roads which was primarily ascribed to the enhanced alertness of the drivers during driving under such situation. It was asserted that the consistency in the horizontal curve along the road can be used as a key determinant factor to ensure the safety of the road users. It was argued that in the presence of poor road horizontal alignment, the drivers were able to control their behaviour and reactions accordingly for maintaining the safety on the road. Therefore, the estimation obtained from the model may be useful to undertake safety measures for lowering the road accident severity in Malaysia. For instance, the drivers must be warned through definite traffic safety awareness programmes regarding the possiblities of fatalities and accidents due to careless driving in the areas having standard curved road segments. However, to disseminate such awareness, intensive efforts from the multiple agencies such as the transport, police, health care, education are mandatory. On top, various proactive measures addressing the driving safety issues are prerequisite in Malaysia as suggested in the Eleventh Malaysia Plan [72].

It worth mentioning that Malaysia has high car ownership due to excessive dependence on the private vehicles. Thus, more interactions on the roads implies higher accident likelihood of the users. Thus, an improvement of the public transportation systems would certainly be useful towards the socio-economic conditions. Increased public transport will not only lessen the traffic congestion and accident exposure but will provide more rational choices for road users, creating a sustainable national transportation for the future. It can be further argued that the expansion of more road infrastructure are not sustainable in the long term. In fact, wide expansion of the road capacity to cater the higher demands of the vehicles only ease the interim problem.

The results obtained from the model revealed that by optimizing the usage of the available personnel and resources an effective strategy for the accidents (with high-risk factors) prevention can be developed. Although the present study covered the available data recorded during 2008 and 2015 only, however the long-term accident data with finer details can be used in the proposed model to predict more accurately the road accident fatalities in Malaysia. Other related information such as the road density, road network, level of service (LOS) of the road facilities, population distribution, socio-economic situation, topography, temporal and spatial factors, and environmental conditions (air, climate change, ambient temperature and noise) must be considered to determine the significant factors that contribute to the higher chances of fatalities. In addition, the impact of the drivers behaviour or characteristics such as the speed choice, age, gender and safety precautions while driving (for example the use of seat belt or helmet) on the accident severity (such as the PDO, injury, or fatality levels) are worth exploring. Inclusion of all these factors may enable us to develop a holistic approach towards the road safety measures. On top, the information from the hospital on the road accidents related injuries can be beneficial to integrate with the PWD and police-related database for developing a complete model. The future work in accident severity model is extended to include the existence of multiple accident severity with bigger sample size, and the use of advanced methodological alternatives that take heterogeous effects into consideration. In short, the proposed accident severity model based disclosure is established to be prospective to improve its predictive ability so that an effective mitigation strategy can be initiated in a holistic way.

## Supporting information

**S1 Data.**
(CSV)

## Author Contributions

**Conceptualization:** Sitti Asmah Hassan, Nordiana Mashros.

**Data curation:** Mohad Fedder Musa.

**Formal analysis:** Mohad Fedder Musa, Sitti Asmah Hassan.

**Funding acquisition:** Sitti Asmah Hassan.

**Investigation:** Mohad Fedder Musa, Sitti Asmah Hassan.

**Methodology:** Mohad Fedder Musa, Sitti Asmah Hassan.

**Resources:** Nordiana Mashros.

**Supervision:** Sitti Asmah Hassan.

**Validation:** Sitti Asmah Hassan.

**Writing – original draft:** Mohad Fedder Musa.

**Writing – review & editing:** Sitti Asmah Hassan.

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
