## [Decision Letter · Decision Letter 0]

7 Apr 2020

PONE-D-20-07171

The impact of roadway conditions on accident severity on Federal Roads in Malaysia

PLOS ONE

Dear Dr Hassan,

Thank you for submitting your manuscript to PLOS ONE. After careful consideration, we feel that it has merit but does not fully meet PLOS ONE’s publication criteria as it currently stands. Therefore, we invite you to submit a revised version of the manuscript that addresses the points raised during the review process.

We would appreciate receiving your revised manuscript by May 22 2020 11:59PM. To enhance the reproducibility of your results, we recommend that if applicable you deposit your laboratory protocols in protocols.io, where a protocol can be assigned its own identifier (DOI) such that it can be cited independently in the future. For instructions see: http://journals.plos.org/plosone/s/submission-guidelines#loc-laboratory-protocols

We look forward to receiving your revised manuscript.

Kind regards,

Feng Chen

Academic Editor

PLOS ONE

Journal Requirements:

3. We note that Figure 1 in your submission contain map/satellite images which may be copyrighted. All PLOS content is published under the Creative Commons Attribution License (CC BY 4.0), which means that the manuscript, images, and Supporting Information files will be freely available online, and any third party is permitted to access, download, copy, distribute, and use these materials in any way, even commercially, with proper attribution. For these reasons, we cannot publish previously copyrighted maps or satellite images created using proprietary data, such as Google software (Google Maps, Street View, and Earth). For more information, see our copyright guidelines: http://journals.plos.org/plosone/s/licenses-and-copyright.

Reviewers' comments:

Reviewer's Responses to Questions

**Comments to the Author**

1. Is the manuscript technically sound, and do the data support the conclusions?

Reviewer #1: Partly

Reviewer #2: No

2. Has the statistical analysis been performed appropriately and rigorously? 

Reviewer #1: No

Reviewer #2: No

3. Have the authors made all data underlying the findings in their manuscript fully available?

Reviewer #1: Yes

Reviewer #2: Yes

4. Is the manuscript presented in an intelligible fashion and written in standard English?

Reviewer #1: No

Reviewer #2: No

5. Review Comments to the Author

Reviewer #1: This paper investigates the effects of roadway conditions on traffic crash severity on Federal roads in Malaysia. Although the authors have found something, the contributions of the findings are very limited. There are several reasons for the limitations of this research: 1. Only the fatal crash data are used in the analysis which cannot provide a comprehensive understanding of crash severity (usually including other severity levels such as no injury, minor injury, and severe injury). Focusing on fatal crashes also significantly reduces the size of crash dataset, which may result in small size problem on model parameter estimation. 2. The binary Logistic regression is too simple, which cannot guarantee the precise of the estimation results. Please refer to the Mannering and Bhat (2014) and Savolainen et al. (2011) for the introduction of more advanced methods. 3. The factors related to drivers, vehicles, and weather conditions which have been found to significantly impact crash severity, are not controlled in the analysis, which may also lead to biased inference. At least, major revisions are required; otherwise, I do not recommend its publication. Some other comments are as follows:

As the current research focuses on analyzing crash severity, the review part of methods for crash frequency should be removed from the Introduction section. Instead, more introduction on the crash severity models should be added.

According to information in the data collection section, there are about 900 serious or minor injury crashes collected. Why are not they used in the analysis of crash severity?

In Table 1, the definitions of good and bad conditions of each variable should be described. They are very vague to readers.

In Table 3, the authors seems to copy the results from the SPSS immediately. The terms, such as “B” and “S.E.”, are not formal.

Mannering, F. L., Bhat C. R., 2014. Analytic methods in accident research: Methodological frontier and future directions. Analytic Methods in Accident Research 1: 1-22.

Savolainen, P. T., Mannering, F. L., Lord, D., Quddus, M. A., 2011. The statistical analysis of highway crash-injury severities: A review and assessment of methodological alternatives. Accident Analysis and Prevention 43 (5), 1666-1676.

Reviewer #2: This paper claims to investigate the impact of roadway conditions on traffic accident severity of federal road in Malaysia. There are some serious flaws with the manuscript as listed below:

1. The data adopted in this paper only included 166 accidents with 1067 cases. This raises several concerns about the validity of the paper. First of all, an investigation towards injury severity using only 166 accidents suffers greatly from the small sample size issues. By processing 166 accidents into 1067 cases makes it even worse. Since it means one accident is split into multiple cases where the severity outcomes will be different but the nine explanatory variables remains the same.

2. The simple logistic regression is not appropriate for analyzing injury severity because it doesn't not account for unobserved heterogeneity issues often present in crash data.

6. PLOS authors have the option to publish the peer review history of their article (what does this mean?). If published, this will include your full peer review and any attached files.

Reviewer #1: No

Reviewer #2: No

---

## [Author Response · Author response to Decision Letter 0]

22 May 2020

1. Figure 1 was drawn by authors based on data reported by the refer Malaysian Automotive Association. Malaysia Automotive Info “Summary of Sales & Production 

Data.” Malaysia; 2017 (reference no. 10). Citation was added to the caption of the figure.

2. I believe the comment was made for Figure 2. Figure 2 contains map image. The figure was removed from the manuscript. Caption of Figure 2 is as below:

Fig 2. Federal roads involve in this particular study was indicated in red line (Source: Public Work Department)

Response to Reviewer 1:

1. Thanks for your kind observation, critics and comments in the overall improvement of the manuscript. 

Road accident fatalities have become top priorities and concerns for Malaysia policymakers, hence, understanding the principal factors that explain accident fatality is considered to be the first step towards the adequate design of an accident prevention strategy related to road condition. This was the original motivation in the previous analysis. 

Based on the comments received, the new data analysis was performed by including all accident severity (no injury, slight injury, serious injury and fatal injury).

We admit the small sample size issues. However, the raw data obtained was from Royal Malaysia Police and Public Work Department database (from 2008 to 2015). 

2. The initial motivation of the study was to adopt the logistic regression approach based on the availability & suitability of data source/type (Reference: Al-Ghamdi AS. Using logistic regression to estimate the influence of accident factors on accident severity. Accident Analysis and Prevention. 2002;34(6):729–741.). 

Based on your suggestion, we re-evaluate the propose model in the revised manuscript by utilizing Ordered Logit Model (OLM)- Mannering and Bhat (2014) and Savolainen et al. (2011). Additional related references were also included.

3. We admit that the accident may due to several factors. The data collection was obtained from a traditional data source in which it is limited in many ways. The Royal Malaysia Police and Public Work Department (PWD) database revealed the accident severity and variables associated to it mostly on road conditions. For instance, the roadway design and maintenance is under the job scope of traffic engineer. Therefore, explanatory variables related to roadway conditions is worth to be explored in Malaysia condition for safety mitigation strategies at practitioners’ levels.

4. Other Comments:

• Methods for crash frequency models have been removed from the manuscript.

• Alternatively, crash severity models have been included in the revised manuscript.

• Understanding the fatal contributing factors was the initial motivation of the study. However, based on your recommendation, the revised version was made to include all accident severity.

• This statement was made before the appearance of Table 1: “The good and poor characteristics of these variables were classified when they conformed to the PWD specification and not in conformity, respectively.” Reference related to the PWD specification was also added at the end of the statement in the revised manuscript.

• “B” and “SE” were omitted from the manuscript.

Response to Reviewer 2:

1. Thanks for your kind observation, critics and comments. We have provided the answer of the same issue mentioned by reviewer #1. 

 Road accident fatalities have become top priorities and concerns for Malaysia policymakers, hence, understanding the principal factors that 

 explain accident fatality is considered to be the first step towards the adequate design of an accident prevention strategy related to road 

 condition. This was the original motivation in the previous analysis. We admit the small sample size issues, but this was the original data 

 obtained from Royal Police Malaysia and Public Work Department database. 

 Given the limitation in data, accident severity model in this study can be used at practitioners’ level in which all the explanatory variables are 

 within their authority.

 Due to small data size, we are unable to consider unobserved heterogeneity factors. According to Mannering et al., 2016, random parameters, 

 latent class, and other unobserved heterogeneity approaches will mitigate the adverse impacts of omitting significant explanatory variables, 

 the resulting model estimates will not be able to track the unobserved heterogeneity as well as when having the significant omitted variables 

 included in the specification. Thus leaving out important explanatory variables still remains a problem even with advanced approaches to 

 account for unobserved heterogeneity. 

 However, statistical approaches that address unobserved heterogeneity tend to be more complex from a model estimation perspective 

 (Reference: Mannering, F. L., Shankar, V., & Bhat, C. R. (2016). Unobserved heterogeneity and the statistical analysis of highway accident 

 data. Analytic Methods in Accident Research, 11, 1–16)

2. Based on your input, the initial binary logistic regression model is now replaced by ordered logistic regression model. All accident severities 

 were evaluated in the new analysis (no injury, slight injury, serious injury, and fatal).

---

## [Decision Letter · Decision Letter 1]

1 Jun 2020

PONE-D-20-07171R1

The impact of roadway conditions towards accident severity on Federal Roads in Malaysia

PLOS ONE

Dear Dr. Hassan,

Thank you for submitting your manuscript to PLOS ONE. After careful consideration, we feel that it has merit but does not fully meet PLOS ONE’s publication criteria as it currently stands. Therefore, we invite you to submit a revised version of the manuscript that addresses the points raised during the review process.

We look forward to receiving your revised manuscript.

Kind regards,

Feng Chen

Academic Editor

PLOS ONE

Reviewers' comments:

Reviewer's Responses to Questions

**Comments to the Author**

1. If the authors have adequately addressed your comments raised in a previous round of review and you feel that this manuscript is now acceptable for publication, you may indicate that here to bypass the “Comments to the Author” section, enter your conflict of interest statement in the “Confidential to Editor” section, and submit your "Accept" recommendation.

Reviewer #1: (No Response)

Reviewer #2: All comments have been addressed

2. Is the manuscript technically sound, and do the data support the conclusions?

Reviewer #1: (No Response)

Reviewer #2: Yes

3. Has the statistical analysis been performed appropriately and rigorously? 

Reviewer #1: (No Response)

Reviewer #2: Yes

4. Have the authors made all data underlying the findings in their manuscript fully available?

Reviewer #1: (No Response)

Reviewer #2: Yes

5. Is the manuscript presented in an intelligible fashion and written in standard English?

Reviewer #1: (No Response)

Reviewer #2: Yes

6. Review Comments to the Author

Reviewer #1: The authors should be thanked for their great efforts on improving the manuscript. Most of my comments have been addressed properly. A minor suggestion is that more works on generalized ordered logit models should be acknowledged in the Introduction Section, including:

Investigating the impacts of real-time weather conditions on freeway crash severity: A Bayesian spatial analysis. International Journal of Environmental Research and Public Health, 2020, 17(8), 2768.

Analyzing freeway crash severity using a Bayesian spatial generalized ordered logit model with conditional autoregressive priors. Accident Analysis and Prevention, 2019, 127, 87-95.

Reviewer #2: The paper has addressed most of the concerns about the paper, however, the following relevant papers should be discussed in the literature review and included in the reference:

[1] Feng Chen, Mingtao Song and Xiaoxiang Ma, Investigation on the Injury Severity of Drivers in Rear-End Collisions Between Cars Using a Random Parameters Bivariate Ordered Probit Model, International Journal of Environmental Research and Public Health, 2019, 16(14) , 2632.

[2] Chen, Feng; Chen, Suren; Ma, Xiaoxiang. Analysis of hourly crash likelihood using unbalanced panel data mixed logit model and real-time driving environmental big data. 2018, JOURNAL OF SAFETY RESEARCH. 65: 153-159.

[3] Bowen Dong, Xiaoxiang Ma, Feng Chen and Suren Chen. “Investigating the Differences of Single- and Multi-vehicle Accident Probability Using Mixed Logit Model", Journal of Advanced Transportation, 2018, UNSP 2702360.

[4] Ma, X., Chen, F., Chen, S., 2015. Empirical analysis of crash injury severity on mountainous and non-mountainous interstate highways. Traffic Inj. Prev. 16 7 , 715–723. doi:10.1080/15389588.2015.1010721

7. PLOS authors have the option to publish the peer review history of their article (what does this mean?). If published, this will include your full peer review and any attached files.

Reviewer #1: No

Reviewer #2: No

---

## [Author Response · Author response to Decision Letter 1]

2 Jun 2020

Response to Reviewer #1:

Thanks for your suggestions.

The additional works on the GLOM have been added in the Introduction section.

Response to Reviewer #2:

Thanks for your suggestions.

The additional references have been added in the manuscript.

---

## [Decision Letter · Decision Letter 2]

18 Jun 2020

The impact of roadway conditions towards accident severity on Federal Roads in Malaysia

PONE-D-20-07171R2

Dear Dr. Hassan,

We’re pleased to inform you that your manuscript has been judged scientifically suitable for publication and will be formally accepted for publication once it meets all outstanding technical requirements.

Kind regards,

Feng Chen

Academic Editor

PLOS ONE

Additional Editor Comments (optional):

Reviewers' comments:

Reviewer's Responses to Questions

**Comments to the Author**

1. If the authors have adequately addressed your comments raised in a previous round of review and you feel that this manuscript is now acceptable for publication, you may indicate that here to bypass the “Comments to the Author” section, enter your conflict of interest statement in the “Confidential to Editor” section, and submit your "Accept" recommendation.

Reviewer #1: All comments have been addressed

Reviewer #2: All comments have been addressed

2. Is the manuscript technically sound, and do the data support the conclusions?

Reviewer #1: (No Response)

Reviewer #2: Yes

3. Has the statistical analysis been performed appropriately and rigorously? 

Reviewer #1: (No Response)

Reviewer #2: Yes

4. Have the authors made all data underlying the findings in their manuscript fully available?

Reviewer #1: (No Response)

Reviewer #2: Yes

5. Is the manuscript presented in an intelligible fashion and written in standard English?

Reviewer #1: (No Response)

Reviewer #2: Yes

6. Review Comments to the Author

Reviewer #1: (No Response)

Reviewer #2: (No Response)

7. PLOS authors have the option to publish the peer review history of their article (what does this mean?). If published, this will include your full peer review and any attached files.

Reviewer #1: No

Reviewer #2: No

---

## [Editor Report · Acceptance letter]

23 Jun 2020

PONE-D-20-07171R2 

The impact of roadway conditions towards accident severity on Federal Roads in Malaysia 

Dear Dr. Hassan:

I'm pleased to inform you that your manuscript has been deemed suitable for publication in PLOS ONE. Congratulations! Your manuscript is now with our production department. 

Kind regards, 

on behalf of

Dr. Feng Chen 

Academic Editor

PLOS ONE